# FOUNDATION VISION MODELS ARE UNSUPERVISED IMAGE CANONICALIZERS

## ABSTRACT

One of the most significant and longstanding problems in computer vision is invariance - the ability to robustly handle changes in real-world transformations such as rotation, viewpoint, and lighting. Unfortunately, popular foundation models remain brittle under such transformations. While existing solutions towards invariance have shown promise, they all fundamentally require some model training, limiting their ability to adapt broadly to new tasks, transformations, and datasets. Our key insight is that foundation model priors can be used to reason about transformations. We thus propose Foundation Model Canonicalization (FMC), an approach that can undo nuisance transformations in images without any model training. With a single core approach, FMC can make models like CLIP and SAM invariant to different transformations without any training or fine-tuning. Our approach FMC flexibly adapts to new foundation models and tasks, making it significantly easier for newer and larger models to achieve invariance.

## 1 INTRODUCTION

One of the oldest and most important problems in vision is *invariance*, and one of the oldest and most promising solutions to invariance is *canonicalization* (Pitts & McCulloch, 1947; Marr & Nishihara, 1978; Hinton, 1981; Tarr & Pinker, 1989; Olshausen et al., 1993). No object appears the same way twice due to rotation, lighting, and viewpoint changes. Yet, the brain can flexibly handle these variations. One leading hypothesis for this impressive ability is that the brain transforms the input into a *canonical* version, thus eliminating the nuisance variations (Shepard & Metzler, 1971).

In contrast, recent foundation models are brittle against such transformations despite their large-scale training (Bommasani et al., 2021). This limitation hurts their applicability to real-world settings because mobile agents frequently encounter objects in unfamiliar contexts (e.g., a robot seeing a chair from an unusual viewpoint). While these models show impressive generalization in-distribution, they are unable to handle *out-of-distribution* inputs.

Achieving invariance in foundation models is especially challenging due to their scale. Re-training or fine-tuning such models can be prohibitively costly or even impossible in many domains. Thus, popular invariance approaches such as data augmentation (Bouchacourt et al., 2021), equivariant architectures (Weiler & Cesa, 2019), or learned invariance (Benton et al., 2020) can be impractical since these approaches rely on re-training or fine-tuning. Even if such a model could be fine-tuned, this procedure reduces the out-of-the-box flexibility and generality of the target foundation model.

Thus, we ask: *Could a single invariance method work for different tasks, models, and transforms?*

Canonicalization is a promising approach because it decouples the invariance method from the downstream model. Once an input has been made "upright", any downstream model can use it. A recent line of work (Mondal et al., 2023; Kaba et al., 2022) shows impressive results across domains through *learned canonicalization*. Specifically, PRLC (Mondal et al., 2023) trains transformation and model-specific canonicalization networks, relying on fine-tuning or dataset-dependent priors for alignment with the downstream model. Still, a key limitation remains: PRLC still requires training specialist networks for each dataset and transformation pair.

We thus propose Foundation Model Canonicalization (FMC), a general canonicalization approach that provides invariance *without any model training* (Figure 1). Our key insight is that foundation

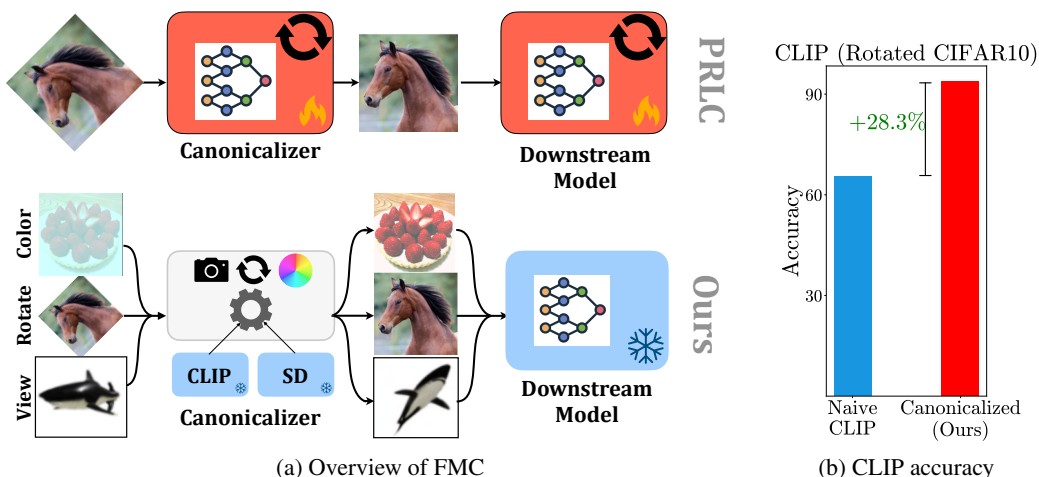

(a) Overview of FMC

(b) CLIP accuracy

Figure 1: **Foundation Model Canonicalization is a training-free method for canonicalization-based invariance**. **(a)** Existing canonicalizers like PRLC (Mondal et al., 2023) train a model-and-task-specific canonicalization network as well as fine-tune the downstream model for best performance. In contrast, we propose a fully unsupervised canonicalizer, FMC, which leverages priors from foundation models like CLIP and Stable Diffusion to canonicalize images for various transforms and downstream models. **(b)** CLIP accuracy drops significantly under rotations. However, applying FMC improves the accuracy under rotations by **28.3%**. In summary, FMC can canonicalize images across different transformations *without training* and make downstream models robust.

model priors about images can be used to reason about transformations and thus perform canonicalization. Specifically, we design energy functions from CLIP (Radford et al., 2021), SAM (Kirillov et al., 2023), and Stable Diffusion (Rombach et al., 2021) to determine which transformation parameters is most likely correspondent to the canonical form (Section 3).

Since FMC does not require training, it can be applied to new foundation models and tasks freely. We first show that FMC *genaralizes across datasets*. We evaluate FMC on CLIP, finding that FMC outperforms PRLC-trained models by at least 16% on pose accuracy (Section 4.1). FMC's superior canonicalization ability enables it to improve accuracy on rotated images, even beating PRLC on settings they specifically trained their canonicalizer and downstream models on whereas FMC is fully unsupervised. On CLIP, we outperform PRLC by 7.4% on CIFAR10, 9.6% on CIFAR100, 2.1% on STL10, and 4.4% on ImageNet. We then show that FMC *generalizes across models*. On top of extending well to CLIP, we find that FMC outperforms PRLC by 26.2% on pose accuracy on SAM Section 4.2. Finally, we also *generalize across transformations*, showing that FMC can canonicalize color chrominance shifts and can improve the accuracy on poor 3D viewpoints.

In summary, this paper aims to provide a general canonicalization approach for invariance at a foundation model scale. **(1)** We propose FMC, an energy function-based approach to canonicalization that achieves invariance without any model training (Section 3). **(2)** Because FMC does not require model training, FMC is much more general than prior works, enabling us to achieve invariance on large scale models like CLIP (Section 4.1) and SAM (Section 4.2). **(3)** We show that FMC can canonicalize other transformations such as color chrominance and 3D viewpoints (Section 4.3).

## 2 BACKGROUND

We start by explaining invariance and canonicalization in terms of group theory as introduced by Kaba et al. (2022)We then explain energy-based models and how they can be extracted from foundation models such as CLIP (Radford et al., 2021) and Stable Diffusion (Rombach et al., 2021).

**Equivariance**: We start with a function $f : \mathcal{X} \to \mathcal{Y}$ with inputs $x \in \mathcal{X}$ and outputs $y \in \mathcal{Y}$. We also assume a group of transformations $G$ acting on the input. Specifically, we denote the transformation as $\mathcal{T}_g : \mathcal{X} \to \mathcal{X}$ where $g \in G$ is a group element. Please note that while Kaba et al. (2022) consider linear symmetries, i.e., $T \in GL(\mathcal{X})$, we consider any transforms $\mathcal{T}_g$ parameterized by a group.

The goal is then to make $\boldsymbol{f}$ *equivariant* or *invariant* to the transformation group. Formally, $\boldsymbol{f}$ is equivariant if:

$$\boldsymbol{f}(\mathcal{T}_g(\boldsymbol{x})) = T'_g \boldsymbol{f}(\boldsymbol{x}) \ \ \forall \boldsymbol{x} \in \mathcal{X}$$

where $T'_g : \mathcal{Y} \to \mathcal{Y}$ is a transform acting on the output space $\mathcal{Y}$. Intuitively, equivariance means that the function's output changes in the same way as the input under the group of transformations. If $T'_g$ is the identity, then $\boldsymbol{f}$ is invariant to the group: $\boldsymbol{f}(\mathcal{T}_g(\boldsymbol{x})) = \boldsymbol{f}(\boldsymbol{x}) \ \ \forall \boldsymbol{x} \in \mathcal{X}$

**Canonicalization**: Canonicalization refers to the process of transforming an input into a canonical version. In the context of images, this could mean rotating an image to be upright or normalizing the lighting. Kaba et al. (2022) formalize canonicalization as a method of equivariance by using a canonicalizer function $\boldsymbol{h} : \mathcal{X} \to \mathcal{T}_g$ that maps the input to a transformation. They write the canonicalized form $\phi$ as:

$$\boldsymbol{\phi}(\boldsymbol{x}) = h'(\boldsymbol{x}) \boldsymbol{f}\big(h(x)\,\boldsymbol{x}\big)$$

where $h(x)$ undoes the transformation on $x$, effectively "uprighting" it, and $h'(x)$ re-applies the transformation to the output. Kaba et al. (2022) show that if the canonicalizer $h$ is defined as a minimizer over transformations, then $\phi$ is guaranteed to be equivariant:

$$h(\boldsymbol{x}) = \underset{\mathcal{T}_g,\ g \in G}{\arg\min}\, E(\mathcal{T}_g(\boldsymbol{x})) \tag{1}$$

where $E : \mathcal{X} \to \mathbb{R}$ is a real-valued function. Strikingly, this holds even when $E$ is not equivariant.

**Energy-based models and energy functions**: EBMs are a class of probabilistic models inspired by statistical mechanics. Given a random variable $\mathrm{x} \in \mathbb{R}^D$, any probability distribution $P_\theta(\mathrm{x})$ can be re-written as:

$$P_\theta(x) = \frac{1}{Z(\theta)} e^{-E_\theta(x)}$$

where $Z(\theta)$ is the normalizing constant and $E_\theta : \mathbb{R}^D \to \mathbb{R}$ is called the energy function. Here, small values of $E_\theta(x)$ correspond to more likely $x$.

These models are especially powerful because multiple EBMs can be composed by combining their corresponding energy functions (Du et al., 2023; Liu et al., 2022) through operations such as addition. In this work, we derive energy functions from models such as CLIP (Radford et al., 2021) and Stable Diffusion (Rombach et al., 2021) and combine them for better canonicalization.

**Classifiers as energy-based models:** Grathwohl et al. (2019) note that any classifier can be seen as an energy-based model using its logits to define the joint energy between input and output. Specifically, the classifier energy for $f_\theta$ with input $x$ and output $y$ is the negative logit:

$$E_\theta(x, y) = -f_\theta(x)[y]$$

and the energy of an input $x$ can be defined using the joint energy $E_\theta(x, y)$ by computing the log-sum-exp over all labels: $E_\theta(x) = -\operatorname{LogSumExp}_y(f_\theta(x)[y])$

While Grathwohl et al. (2019) use this to define a trainable energy-based model, we use a *pre-trained* classifier (specifically CLIP) and find that the energy function derived from it can be used effectively for canonicalization. We also replaced $\operatorname{LSE}(.)$ with $\max(.)$ for simplicity.

**Diffusion models as priors**: Graikos et al. (2022) show that diffusion models perform well as image priors. Specifically, they perform inference over the data distribution $x \sim p(x)$ with a differentiable constraint $c(x, y)$ where $y$ is some additional information. The task of modeling the desired posterior $p(x|y)$ is modeled as a minimization of free energy, as presented in Graikos et al. (2022):

$$E_\theta(\boldsymbol{\eta}) = \sum_t \mathbb{E}_{\boldsymbol{\epsilon} \sim \mathcal{N}(\boldsymbol{0}, \mathbf{I})} \left[ ||\boldsymbol{\epsilon} - \boldsymbol{\epsilon}_\theta(\mathbf{x_t}, t)||_2^2 \right] - \log c(\boldsymbol{\eta}, \boldsymbol{y}), \quad \mathbf{x_t} = \sqrt{\bar{\alpha}_t}\boldsymbol{\eta} + \sqrt{1 - \bar{\alpha}_t}\boldsymbol{\epsilon} \tag{2}$$

where $t$ goes over the number of diffusion steps, $\boldsymbol{\epsilon_\theta}$ is the pre-trained diffusion model, $\boldsymbol{x_t}$ is the input at time step $t$, $\eta$ is the noise, and $\bar{\alpha}_t = \prod_{i=1}^t (1 - \beta_t)$ where $\beta_t$ is the denoising schedule parameter.

Similarly, Li et al. (2023) use diffusion models as classifiers by minimizing a similar energy function. We use this free energy (without any constraint function) to canonicalize inputs. Specifically, we will show how minimizing this free energy over rotations or lighting correlates to upright images. This can help achieve equivariance with a pre-trained model such as CLIP (Radford et al. (2021)).

Figure 2: **Transformation distributions define a slice through the distribution of natural images, enabling us to use foundation models to canonicalize.** Given the complex distribution of natural data, which spans across many transformations, we propose a common solution that applies across arbitrary transformation-based slices of the distribution (left). Given a particular slice, we simulate different versions of the input image along this slice of the distribution, using energy functions built on CLIP (Radford et al., 2021) and Stable Diffusion (Rombach et al., 2021) models for each sample. We minimize the total energy to predict the canonical version (right).

## 3 METHODS

We start with our key insight and describe the overall framework of FMC (Section 3.1). We then define the energy functions based on each foundation model (CLIP, Stable Diffusion, SAM) and how to combine them (Section 3.2). Finally, we then describe how we use Bayesian Optimization to efficiently search the continuous space of transformations (Section 3.3).

### 3.1 KEY INSIGHT AND FRAMEWORK

**Key insight**: An image $x$, along with all of its transformed versions $\mathcal{T}_g(x)$, defines a *slice* through the overall distribution of natural images (Figure 2). In this slice, the canonical version of the image, $x^*$ is likely to be encountered the most often in training data (similar to the upright assumption of PRLC (Mondal et al., 2023)). Formally:

$$p_{\text{data}}(x^*) \geq p_{\text{data}}(\mathcal{T}_g(x^*)) \quad \forall g \in G$$

Maximizing this probability over all transformations is sufficient to find the canonical $x^*$. A general model of the natural image distribution might thus be used to canonicalize images across a large range of transformations.

**Framework:** We define an energy function $E_{\text{FMC}} \propto -\log p_{\text{data}}(x)$ estimated by foundation models to be used as the canonicalizer in Equation (1). Minimizing this energy over transformations thus maximizes the probability of the resulting image, thus canonicalizing it:

$$y = f(\mathcal{T}_{\hat{g}}(x)) \quad \text{where} \quad \hat{g} = \arg\min_{g \in G} E_{\text{FMC}}(\mathcal{T}_g(x)) \tag{3}$$

Our framework is thus divided into three components: (1) A parametrized transformation $\mathcal{T}_g$ (e.g., rotation), (2) An energy function $E_{\text{FMC}}$ derived from various foundation models, and (3) A downstream model $f$ that performs the desired task.

Please note that this framework is fully modular, with the necessary equivariance/invariance *emerging* from the system design rather than an inherent property of any single component.

### 3.2 ENERGY FUNCTIONS FROM FOUNDATION MODELS

We take existing foundation models such as CLIP, Stable Diffusion, and SAM and extract knowledge from them in the form of energy functions. This is an especially convenient form of knowledge since the energy functions from multiple models can be readily combined with each other as well as hand-designed priors. In particular, each energy is defined as:

1. $E_{\text{uncond}}(x; \alpha, \beta)$: Following joint energy models (JEM) (Grathwohl et al., 2019), the unconditional classifier energy marginalizes over all classes, thus not requiring a class label. However, it is defined in a simpler manner as a linear combination of mean and max logits:

$$E_{\text{uncond}}(x; \alpha, \beta) = \alpha \frac{1}{|C|} \sum_{c=1}^{|C|} f_\theta(x)[c] - \beta \max_{c \in 1,2,\ldots,|C|} f_\theta(x)[c]$$

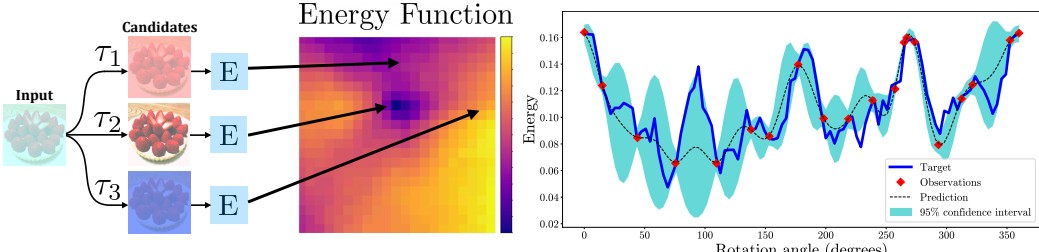

Figure 3: **Foundation Model Canonicalization in continuous transformation spaces**: Given an input image, we generate different transformed versions of the image (left). Each is ranked by a combined energy function as shown in Figure 2 with the minimum of this grid representing the canonical form (center). However, this is infeasible for continuous transformation spaces such as color shift. Thus, we apply Bayesian Optimization to estimate the minimum energy in continuous spaces (right). Combining energy functions and Bayesian optimization provides a general approach for canonicalization in continuous transform spaces.

where $\alpha, \beta \in \mathbb{R}$ are hyperparameters and $f_\theta(\boldsymbol{x})[c]$ is the logit for image $\boldsymbol{x}$ and class $c$. For CLIP, we use cosine similarity between the CLIP image embedding of image $\boldsymbol{x}$ and text embedding of label $c$ instead. We use CLIP ViT-H-14 for this energy function.

2. $E_{\text{diff}}(\boldsymbol{x})$: This energy uses an unconditional diffusion model to impose a prior on latent images. Following Graikos et al. (2022), the energy is the negative diffusion model loss:

$$E_{\text{diff}}(\boldsymbol{x}) = \frac{1}{T} \sum_{t=1}^{T} \|\epsilon_t - \epsilon_\theta\big(\sqrt{\bar{\alpha}_t}\boldsymbol{x} + \sqrt{1 - \bar{\alpha}_t}\epsilon_t\big)\|^2$$

where $\boldsymbol{x}$ is the image. Interestingly, we find that using only a subset of time steps $(500 - 1000)$ is sufficient for the transformations considered in this paper. We use Stable-Diffusion-2-base for all our experiments.

3. $E_{\text{seg}}(\boldsymbol{x})$: The Segment Anything Model (SAM) Kirillov et al. (2023) produces an estimated IoU for each candidate segmentation for a given image and prompt (i.e., box, caption, points). While SAM does not explicitly model $p_{\text{data}}$, its IoU is likely to be higher for in-distribution images. It thus can be used as a weak proxy for the likelihood of the observed image. We use the negative estimated IoU (eIoU) from SAM-ViT-H as our energy function:

$$E_{\text{seg}}(\boldsymbol{x}) = -\text{eIoU}_{\text{SAM}}(\boldsymbol{x})$$

**Combining energy functions**: We minimize the combined energy $E_{\text{FMC}}\big(\mathcal{T}_g(\boldsymbol{x})\big)$ over all transformations $\mathcal{T}_g$ to find the canonical version of the input image $\boldsymbol{x}$. This is done by solving the following optimization problem:

$$\hat{g} = \underset{g \in G}{\arg\min}\, E_{\text{FMC}}\big(\mathcal{T}_g(\boldsymbol{x}), \alpha, \beta, \gamma_1, \gamma_2, \gamma_3\big) \tag{4}$$

$$E_{\text{FMC}}\big(\mathcal{T}_g(\boldsymbol{x}), \alpha, \beta, \gamma_1, \gamma_2, \gamma_3\big) = \gamma_1 E_{\text{uncond}}(\mathcal{T}_g(\boldsymbol{x}); \alpha, \beta) + \gamma_2 E_{\text{diff}}(\mathcal{T}_g(\boldsymbol{x})) + \gamma_3 E_{\text{seg}}(\mathcal{T}_g(\boldsymbol{x})) \tag{5}$$

where $\mathcal{T}$ is the set of all transformations and $\alpha, \beta, \gamma_1, \gamma_2, \gamma_3 \in \mathbb{R}$ are hyperparameters.

### 3.3 BAYESIAN OPTIMIZATION FOR CONTINUOUS ENERGY LANDSCAPES

While the energy function in Equation (4) can be minimized through exhaustive search for a small number of transformations (like C8, i.e. the group of 8 rotations around the circle), it becomes infeasible for continuous transformations. Gradient-based optimization requires differentiating through the energy function and, thus, the foundation models, which is infeasible due to drastically higher memory cost. PRLC (Mondal et al., 2023) uses SO(3)-equivariant canonicalization network to predict rotations, but this solution does not work for all continuous domains.

Since this is a low-dimensional continuous optimization problem, we use the well-established Bayesian Optimization method (Nogueira, 2014; Frazier, 2018) to efficiently minimize the energy function with a small number of evaluations. Specifically, we use a Gaussian Process (GP) with an

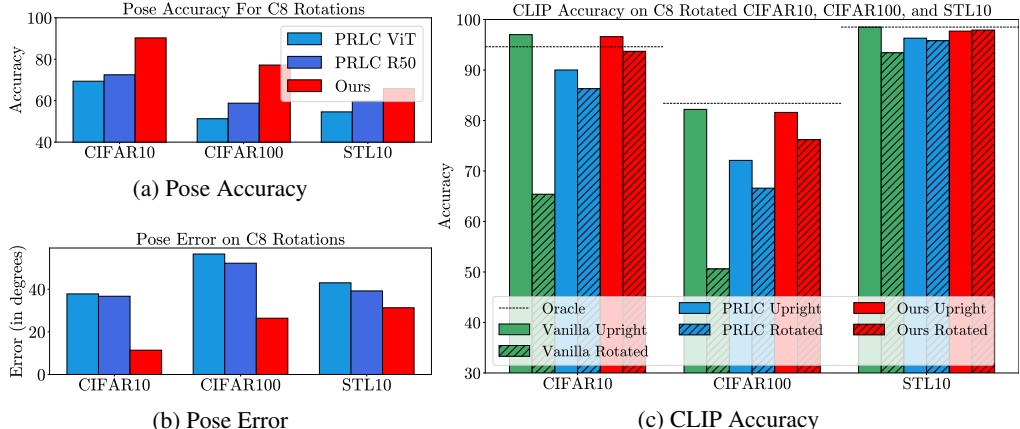

(a) Pose Accuracy

(b) Pose Error

(c) CLIP Accuracy

Figure 4: **FMC beats PRLC dataset specialist canonicalizers on rotated accuracy and pose estimation. (a)** We show that FMC achieves better pose accuracy than PRLC on their datasets. **(b)** We show that FMC achieves model beats PRLC dataset specialist models on pose error. **(c)** As a result of our superior pose estimation, we generalize better to new models like CLIP. Dashed lines represent oracle performance, i.e., perfectly undoing the rotation except for any loss of corner information from cropping. This result highlights FMC's strong canonicalization ability.

RBF kernel to model the energy function from a small number of sample evaluations. We use the GP to guide the search for the optimal transformation $\hat{g}$ using the expected improvement (EI) acquisition function (Jones et al., 1998), which is defined as: $\text{EI}(g) = \mathbb{E}\left[(E_{\min} - \hat{E}_{\text{FMC}}(g)^+\right]$ where $E_{\min}$ is the minimum value of the energy function observed so far, $\hat{E}_{\text{FMC}}$ is the GP's prediction of the energy for $g$, and $(x)^+ = \max(x, 0)$. The $g$ that maximizes the expected improvement is then evaluated and used to update the GP model. The search continues for a fixed number of steps. This allows us to optimize the energy function in continuous spaces.

## 4 EXPERIMENTS

We now evaluate FMC to demonstrate its capabilities as an unsupervised canonicalizer. We first show that FMC generalizes better than PRLC Mondal et al. (2023) across *datasets*, extending to new datasets like ImageNet (Deng et al., 2009) and outperforming the PRLC specialist models trained on CIFAR10 (Krizhevsky et al., 2010), CIFAR100 (Krizhevsky et al., 2010), and STL10 (Coates et al., 2011) (Section 4.1). We then show that FMC generalizes better across *models*, extending to CLIP and outperforming PRLC on SAM (Section 4.2). Finally, we surprisingly find that FMC can generalize to other transformations such as color shifts and 3D viewpoint rotations (Section 4.3).

### 4.1 FMC GENERALIZES ACROSS DATASETS

**Experimental Setup**: We compare against PRLC (Mondal et al., 2023) on their settings on CLIP (Radford et al., 2021), PRLC-trained ViT (Dosovitskiy et al., 2021) PRLC-trained ResNet50 (He et al., 2016) models across CIFAR10 (Krizhevsky et al., 2010), CIFAR100 (Krizhevsky et al., 2010), and STL10 (Coates et al., 2011). For PRLC on CLIP we transfer their ResNet50 canonicalizers. We follow their experimental setup, evaluating on $C8$ rotations. We then extend to ImageNet (Deng et al., 2009) where we use a pretrained ResNet50 and ViT-B. We report accuracy on upright images, rotated images (on $C8$ rotations), pose accuracy (i.e., did the canonicalizer pick the correct $C8$ rotation), and pose error (i.e., the average error in degrees).

**Takeaway #1: FMC Outperforms PRLC Specialists**: Figure 4 shows that FMC beats PRLC dataset specialist canonicalizers on rotated accuracy and $C8$ pose estimation. We find that on each of PRLC's evaluated datasets (CIFAR10, CIFAR100, and STL10) on CLIP, we achieve at least 16% higher $C8$ pose accuracy and at least 7.9% lower pose error. This explains our better generalization to CLIP. We outperform PRLC by 7.4% on CIFAR10, 9.6% on CIFAR100, and 2.1% on STL10. These results highlight FMC's canonicalization ability.

| Pretrained Network | | ResNet50 (PRLC-Trained) | | ViT (PRLC-Trained) | |
|---|---|---|---|---|---|
| Datasets | Canonicalizer | Accuracy | Random Rotation (*C8*) | Accuracy | Random Rotation (*C8*) |
| CIFAR10 | None | 96.6 | 86.2 | 97.6 | 86.7 |
| | Rotation Aug. | 94.9 | 90.1 | 96.3 | 93.7 |
| | PRLC | 96.1 | 95.1 | 95.8 | 94.8 |
| | Ours | **96.4 (+0.3%)** | **95.6 (+0.5%)** | **97.3 (+1.5%)** | **96.0 (+1.2%)** |
| | *Oracle* | *96.6* | *95.9* | *97.6* | *96.7* |
| CIFAR100 | None | 84.4 | 69.7 | 87.1 | 73.8 |
| | Rotation Aug. | 80.2 | 74.1 | 82.6 | 78.4 |
| | PRLC | 83.1 | 81.8 | 83.9 | 82.2 |
| | Ours | **83.7 (+0.6%)** | **82.2 (+0.4%)** | **86.2 (+2.3%)** | **84.3 (+2.1%)** |
| | *Oracle* | *84.4* | *83.4* | *87.1* | *83.4* |
| STL10 | None | 97.4 | 88.7 | 97.3 | 90.0 |
| | Rotation Aug. | 98.1 | 95.0 | 97.9 | 94.1 |
| | PRLC | 95.2 | 94.1 | 95.7 | 93.9 |
| | Ours | **96.1 (+0.9%)** | **95.5 (+1.4%)** | **96.0 (+0.3%)** | **95.2 (+1.3%)** |
| | *Oracle* | *97.4* | *96.7* | *97.3* | *96.4* |

Table 1: **FMC beats PRLC dataset specialist canonicalizers and models on rotated accuracy.** We find that FMC outperforms PRLC, without any training, across all PRLC specific model and dataset pairs on both upright inputs and randomly rotated inputs. We compare against just upright images in the Acc columns. Oracle refers to a system where the exact angle to upright is known, and thus only measures the change in accuracy due to loss of information due to rotating, cropping, and re-rotating. Rand Rot. (*C8*) applies a random *C8* transform to the input before passing it to the aligner / model. Best non-oracle rows are bolded. Rotation Aug. numbers taken from Mondal et al. (2023). This result highlights that we can beat PRLC even on their best settings.

Table 1 shows that we outperform PRLC specialists even when using their trained models as the downstream classifier. These classifiers have been fine-tuned to align with the PRLC canonicalizer. Even though our technique is training-free, we outperform these specific dataset and model pairs on CIFAR10, CIFAR100, and STL10 on rotated $C8$ accuracy. This further highlights FMC's strong canonicalization abilities to achieve invariant predictions.

**Takeaway #2: FMC Generalizes to ImageNet better than PRLC**: Table 3 (in the Appendix) evaluates on ImageNet using vanilla ResNet50 and ViT models on ImageNet since PRLC does not train on ImageNet. We take the best-performing PRLC aligner (STL10) for both models. Our unsupervised approach generalizes to new datasets and outperforms PRLC by 4.3% on ResNet50 for $C8$ images and by 11.4% for ViT. Our pose error is at least 17.8% lower than PRLC. While our pose accuracy is slightly lower, if we compute the accuracy of $\pm45°$, we yet again beat PRLC by 22.5%. This suggests that FMC can more consistently generate close to correct poses. In summary, without any ImageNet training, we are able to exhibit better generalization to ImageNet classification on rotated images than PRLC.

**Takeaway #3: PRLC Struggles to Generalize Across Datasets**: Figure 7 (in the Appendix) shows that struggles to generalize across datasets. Its performance on pose estimation drops significantly when using canonicalizers trained on a different datasets than the downstream model. This is most obvious when applying CIFAR10 or CIFAR100 aligners to STL10, with the pose accuracy dropping over 30% on ViT and the pose error rising by 30%. The exception is that the CIFAR100 aligner performs better on CIFAR10 than the CIFAR10 aligner in most cases, likely due to their similarity. These results showcase FMC's strength in generalizing across datasets (Table 1).

## 4.2 FMC GENERALIZES ACROSS MODELS

**Experimental Setup**: We adopt a similar setup to Section 4.1 and PRLC (Mondal et al., 2023). We compare PRLC against FMC on CLIP for classification and SAM for segmentation. We measure the performance on upright images, rotated images (following PRLC, $C8$ for classification and $C4$ for segmentation), and pose accuracy. We also measure the transferability of both PRLC canonicalizers and FMC over CLIP and PRLC trained classifiers. Please see the appendix for further details.

| Pretrained Network | | SAM | |
|---|---|---|---|
| Dataset | Canonicalizer | mAP Random Rotation ($C4$) (%) | Pose Accuracy (%) |
| COCO | Naive | 62.0 | - |
| | PRLC | 62.7 | 47.9 |
| | Ours | **66.1** (+3.4%) | **74.1** (+26.2%) |
| | *Oracle* | *66.2* | - |

Table 2: **FMC beats PRLC on segmentation**: We first report FMC's performance on mAP on COCO, outperforming PRLC by 3.4% (left). We then show that the $C4$ pose accuracy is higher than PRLC by 26.2%. This shows FMC's ability to generalize to segmentation without supervision.

**Takeaway #4: FMC Generalizes to CLIP better than PRLC**: As shown in Section 4.1, FMC generalizes better to CLIP (Figure 4). For each of PRLC's evaluated dataset, we achieve at least a 16% higher $C8$ pose accuracy with a decrease of at least 7.9% on pose error. FMC's strong canonicalization ability enables us to generalize to new models like CLIP, outperforming PRLC by 7.4% on CIFAR10, 9.6% on CIFAR100, 2.1% on STL10, and 4.4% on ImageNet. These results highlight FMC's generalization to new downstream models.

**Takeaway #5: FMC Outperforms PRLC on SAM**: Table 2 shows performance of FMC on SAM following PRLC's setup (SAM-ViT-H on COCO with ground-truth box prompts). We outperform PRLC (Mondal et al., 2023), achieving a 26.2% improvement on $C4$ pose accuracy. We also achieve a 3.4% gain in accuracy over $C4$ rotations. These results demonstrate FMC's ability to generalize across tasks and models to segmentation.

**Takeaway #6: FMC Generalizes Across Downstream Models**: In Fig. 9(in the Appendix) we apply different model aligners to different downstream models. FMC generalizes across other downstream models better than PRLC (Mondal et al., 2023), showing a particular advantage in generalizing to CLIP over the CIFAR10 and CIFAR100 PRLC canonicalizers. These results show that FMC is not limited to working well on CLIP and can generalize across to other downstream models in ViT and PRLC better than PRLC to CLIP.

**Takeaway #7: FMC Produces Stable Classification Results over Angles**: Fig. 8 (in the Appendix) shows the stability of accuracy vs. angle for CLIP. We find that FMC accuracy remains stable across the range of angles, with the accuracy line remaining above that of PRLC on CIFAR10, CIFAR100, and STL10.

### 4.3 FMC GENERALIZES ACROSS TRANSFORMS

We now examine the ability of FMC's ability to generalize to color and 3D viewpoint shifts. A surprising finding is that the unsupervised nature of FMC leads to the emergence of canonicalization in other transformations. We use the von Kries model (KRIES, 1905) to apply color shifts for 3D viewpoints we apply Zero123 Liu et al. (2023) to generate new viewpoints.

**Takeaway #8: FMC Generalizes to Color Chrominance Shifts**: We test the model's invariance to these transformations by applying the above color shifts to the CIFAR100 dataset and evaluating CLIP with and without canonicalization. Figure 5a shows FMC achieves invariance on color shifts. While our method is not competitive against SOTA supervised approaches like Barron & Tsai (2017); Hernandez-Juarez et al. (2020), it is still good enough for classification. Compared to the vanilla model FMC improves the accuracy on chrominance shifted images by 9.9% without any training or adaptations to handle color chrominance. Figure 5b shows accuracy over the radius in log-chrominance space, demonstrating that FMC's accuracy remains more stable than vanilla CLIP. Overall, these results show the surprising ability of FMC to canonicalize color -shifted images.

**Takeaway #9: FMC Generalizes to 3D** Figure 6a shows that FMC's energy function correlates with quality of 3D viewpoint for classification accuracy. Figure 6b shows a histogram of the Spearman's rank correlation coefficients over videos where we find that for the majority of videos, our energy and the ground truth probability are highly correlated, suggesting that FMC's energy function sort 3D viewpoints well. Fig. 6c shows that FMC can improve poor viewpoints. For the worst 11 viewpoints, the accuracy when taking the best valued Zero123 generated image we can improve the accuracy by up to 11.4% and an average of 8.4%.

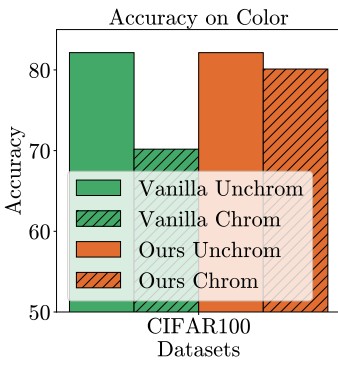

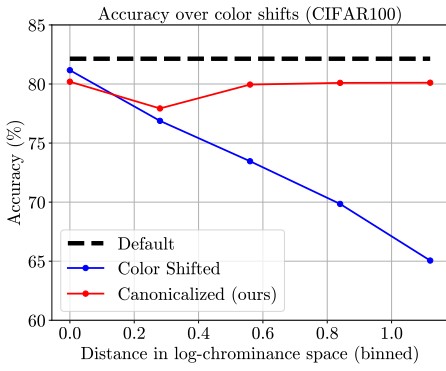

(a) Accuracy over Color Chromi-
nance Shifts.

(b) Accuracy vs Chrominance.

Figure 5: **FMC can canonicalize color chrominance**. **(a)** We show that FMC improves accuracy on chrominance shifted images by 9.9%. **(b)** As the shift distance in log-chrominance space increases, FMC accuracy remains stable while the accuracy of the vanilla model drops.

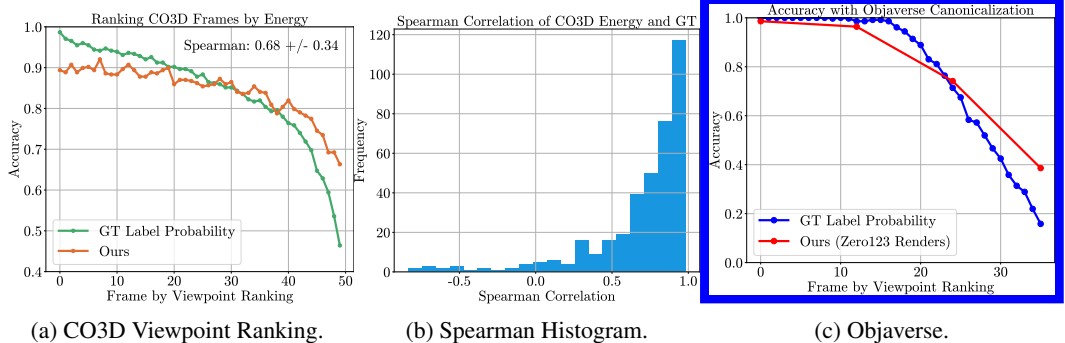

(a) CO3D Viewpoint Ranking.

(b) Spearman Histogram.

(c) Objaverse.

Figure 6: **FMC can improve classification on poor viewpoints**. **(a)** We compare FMC's energy ranking vs. a ground truth ranking of the true label probability on CO3D. **(b)** Histogram of Spearman Correlation coefficients. For most videos, our energy ranking is highly correlated with the ranking generated from using the ground truth label probabilities. **(c)** Objaverse 3D results with Zero123. We compare the accuracy of original frames and the accuracy obtained by taking the Zero123 generated image with the minimum FMC energy. This plot shows that FMC can improve the accuracy on poor viewpoints. These results highlight that FMC can improve classification on poor viewpoints.

## 5 DISCUSSION

**Limitations:** As an unsupervised technique, we do not outperform SOTA on pose estimation benchmarks or color correction methods such as Hernandez-Juarez et al. (2020). For 3D, FMC's ability to classify good viewpoints is weakened for modest gains at worse viewpoints. Further improvements and analysis on how to configure FMC with Zero123 could improve performance. Finally, our technique is slow at inference time due to repeated evaluations of large models. This cost could be further reduced by only using canonicalization for OOD inputs.

**Choice of Foundation Models:** The core principle behind FMC is based on the ability of foundation models to reason about the natural distribution of transformations. We now discuss the requirements for suitable foundation models to provide accurate priors.

In theory, the probability distribution represented by our energy functions (Section 2) must approximate the data distribution along the submanifold defined by the transformation to work well. The suitable foundation models must then: 1) see enough natural data to be able to model the data distribution across multiple settings, 2) have been trained on realistic data that reflects the natural distribution of images rather than augmented data which may distort the learned prior, and 3) employ a training loss that can model the data distribution either explicitly (e.g., flow models) or implicitly (e.g., classifiers through JEM (Grathwohl et al., 2019)). In essence, if the foundation model is a use-

ful out-of-distribution detector for the desired transformations, it is likely to work for FMC. Once foundation models have been selected, the balance between them can be tuned with hyperparameter selection via Bayesian Optimization.

# 6    RELATED WORK

**Data Augmentation**: Data augmentation during training is the simplest and most popular way to achieve invariance, but it requires fixing the transformations ahead-of-time. Some recent work such as VIAT (Ruan et al. (2023)), ViewFool (Dong et al. (2022)), and Omniview Tuning (Ruan et al. (2024)) use adversarial viewpoints as augmentations during training/fine-tuning. However, adapting an existing model to new transformations thus requires expensive re-training or fine-tuning. Additionally, the range of augmentations (e.g., rotation degrees) is unknown and artificially chosen, which can hurt accuracy for some classes (Bouchacourt et al., 2021; Kirichenko et al., 2024). Additionally, the resulting model is not as robust for classes with fewer training examples (Zhou et al., 2022), making data augmentation unsuitable for tasks with imbalanced data.

**Equivariant Networks**: Another line of work (Bronstein et al., 2021) aims to design neural networks with the necessary equivariance hardcoded into the architecture itself. This approach led to Convolutional Neural Networks (LeCun et al., 1999; Fukushima, 1988), and more recently, group equivariant networks for various transforms Cohen et al. (2019); Esteves et al. (2017); Kondor & Trivedi (2018). This elegant approach is useful when the group of transformations is known and fixed (e.g., 2D rotations on images or 3D rotations of point clouds). However, this approach severely limits the choice of architecture. In contrast, our approach does not restrict the underlying models.

**Learned Invariance**: Augerino (Benton et al. (2020)) invariances from the dataset, learning the augmentation range for each transformation independently. LILA (Immer et al. (2022)) improves upon Augerino by using marginal likelihood methods. InstaAug (Miao et al. (2022)) learns per-instance invariances but still struggles with multi-model distributions as it can only model transformation parameters independently. Singhal et al. (2023) generalize Augerino, LILA, and InstaAug by training a normalizing flow to jointly predict transformation distributions for all transformations. This flow-based approach generalizes better and adapts to long-tailed data but still requires training. In contrast, our approach does not require any dataset-specific training. Our key insight is that the knowledge needed to achieve invariance already exists in foundation models, and we propose a way to extract these priors. Our approach thus generalizes better across datasets and can be plug-and-play with classifier dataset pairs without further retraining.

**Learned Canonicalization**: Learned canonicalization has its early roots in mental rotation Shepard & Metzler (1971). Hock & Tromley (1978) found that the response time in humans to recognize rotated objects increased linearly with rotation. Tarr & Pinker (1989) drew further ties between mental rotation and invariant object recognition. These works suggest that canonicalizing can robustly align neural networks to the adaptable nature of the human brain.

Kaba et al. (2022) propose a learned canonicalization (LC) approach via a learned energy function. At test time, it minimizes this function to canonicalize the inputs before passing them through the downstream model, enabling better generalization. EquiAdapt (Mondal et al. (2023)) enforces a regularization prior to align the training set distributions with the predicted canonicalization distributions, improving this approach for downstream models.

However, LC and EquiAdapt still require dataset, transform, and model-specific training and do not generalize well beyond their trained settings (Section 4). In contrast, our approach makes no such assumptions, instead leveraging energy functions of pre-trained foundation models.

We discuss more related work in the Appendix.

**Conclusion**: FMC is a training-free method for canonicalization. We note that foundation model priors can be used to reason about transformations. We can outperform PRLC (Mondal et al., 2023) on their specific settings, generalize better to new datasets and models like ImageNet (Deng et al., 2009) and CLIP (Radford et al., 2021), and extend to transformations other than 2D rotations without any training or fine-tuning. FMC is designed to adapt to new models and tasks, enabling equivariance without the burden of model training.

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

| ImageNet | ResNet50 (Vanilla-Trained) | | ViT (Vanilla-Trained) | | CLIP (Vanilla-Trained) | |
|---|---|---|---|---|---|---|
| Canon. | Acc | Rand Rot. (C8) | Acc | Rand Rot. (C8) | Acc | Rand Rot. (C8) |
| None | 75.2 | 50.1 | 80.4 | 59.6 | 77.1 | 67.0 |
| PRLC* | 63.1 | 59.2 | 63.7 | 60.5 | 72.1 | 69.6 |
| Ours | **66.3** (+3.2) | **63.5** (+4.3) | **73.6** (+9.9) | **71.9** (+11.4) | **75.4** (+3.3) | **74.0** (+4.4) |
| *Oracle* | *75.2* | *71.5* | *80.4* | *78.1* | *77.1* | *75.3* |

| PRLC R50 Aligner | | | PRLC ViT Aligner | | | Ours | | |
|---|---|---|---|---|---|---|---|---|
| Acc (↑) | Acc @ 45° (↑) | Err (↓) | Acc | Acc @ 45° | Err | Acc | Acc @ 45° | Err |
| **37.9** | 55.4 | 63.1 | 31.8 | 56.4 | 65.6 | 37.0 (-0.9) | **78.9** (+22.5) | **45.3** (-17.8) |

Table 3: **FMC generalizes better to ImageNet and outperforms PRLC's canonicalizers.** We find that Foundation Model Canonicalization outperforms PRLC, without any training, on both upright inputs and randomly rotated inputs. We compare against just upright images in the Acc columns. Oracle refers to a system where the exact angle to upright is known, and thus only measures the change in accuracy due to loss of information due to rotating, cropping, and re-rotating. Rand Rot. ($C8$) applies a random $C8$ transform to the input before passing it to the aligner / model. Best non-oracle rows on rotated performance are bolded. For PRLC, the canonicalizers were the best performing ones from other datasets (STL10 for both ResNet50 and ViT). Thus, they were not trained specifically for ImageNet.

# 7 ADDITIONAL RESULTS AND FIGURES

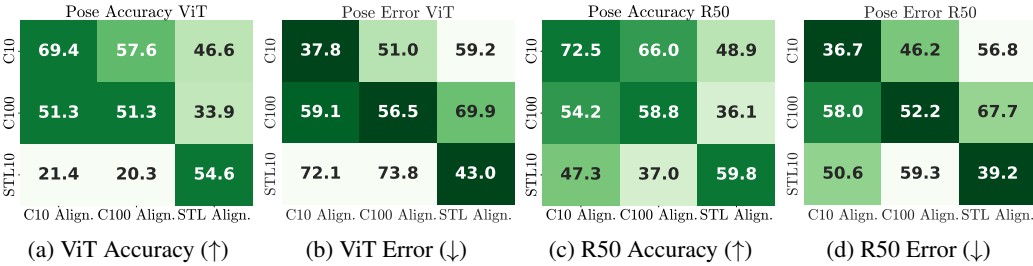

(a) ViT Accuracy (↑)    (b) ViT Error (↓)    (c) R50 Accuracy (↑)    (d) R50 Error (↓)

Figure 7: **FMC generalizes better across datasets when mixing up aligners and downstream models**. PRLC performance on pose estimation drops significantly when using a canonicalizer trained from a different dataset compared to FMC, which applies one technique across all settings. This result highlights the generalizability across datasets of an unsupervised approach.

# A EXPERIMENTAL SETUP

## A.1 EXPERIMENTAL SETUP - 3D

For 3D, we first look at the CO3D (Reizenstein et al., 2021) dataset to measure how our FMC's energy function correlates to 3D viewpoint quality. We compare the ranking of viewpoint frames by FMC energy compared to that of the probability of the ground truth label. We then look at Objaverse-LVIS (Deitke et al., 2022) to measure the effect of combining FMC with Zero123 (Liu et al., 2023) as the transformation generation function to simulate new 3D viewpoints from a single image. For Zero123 experiments, we rank the viewpoints by the probability of the ground truth label and then measure the difference in accuracy between the original Objaverse renders and that of the energy minimizing Zero123 render for the respective ranking bins.

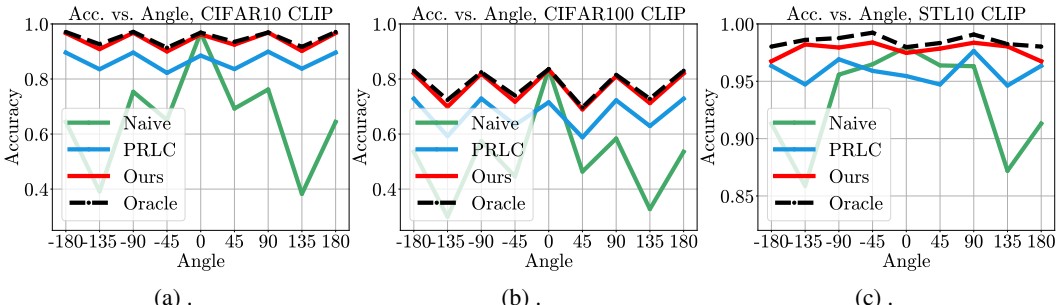

Figure 8: Accuracy vs. $C8$ angle on CLIP. Like on ResNet50, we find that using FMC leads to invariant predictions over angles, outperforming PRLC. The contrast is particularly clear for CLIP on CIFAR10 and CIFAR100, where our accuracy over angle is consistently above PRLC.

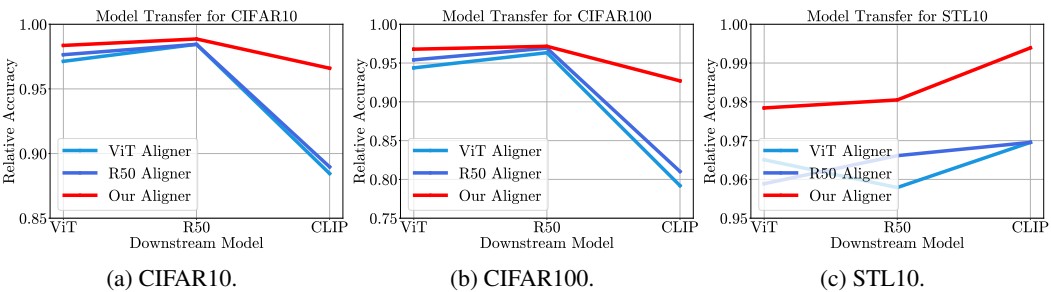

Figure 9: Transferring canonicalizers across models. We measure the effects of transferring canonicalizers across different downstream models by plotting the relative accuracy over the naive model. We find that FMC outperforms the PRLC aligners, particularly when transferring to CLIP on CIFAR10 and CIFAR100. These results show the ability of FMC to generalize across downstream models. All ViT and R50 models and aligners are PRLC-trained versions.

The model we use for both experiments is the fine-tuned version of CLIP from OVSEG (Liang et al., 2023) which is designed to work on background removed images, as Zero123 operates on such images. Please see the Appendix for more details on experimental setup.

## A.2 EXPERIMENTAL SETUP – COLOR

We define the color shift transformation using the popular von Kries model (KRIES, 1905) where an illuminant vector with the RGB values $L = [L_R, L_G, L_B] \in \mathbb{R}^3$ is multiplied element-wise with every pixel in the image. We then generate this illuminant vector $L$ by sampling in the log-chrominance space (Barron & Tsai, 2017). Specifically,

$$L_u, L_v \sim U[-1, 1] \tag{6}$$

$$[L_R, L_G, L_B] = [\frac{\exp(-L_u)}{z}, \frac{1}{z}, \frac{\exp(-L_v)}{z}] \tag{7}$$

where $z = \sqrt{\exp(-L_u)^2 + \exp(-L_v^2) + 1}$ is a normalizing constant and $L_u, L_v$ are the log-chroma values sampled from the uniform distribution with range $[-1, 1]$. Intuitively, the log-chroma space defines the $R/G$ and $B/G$ ratios in log-space. A range of $[-1, 1]$ corresponds roughly to a $7\times$ change in the ratio between the minimum and maximum points of the range.

## A.3 HYPERPARAMETERS FOR THE ENERGY FUNCTIONS

All hyperparameters were found using Bayesian Optimization with the same kernel and acquisition function mentioned in Section 3 and performed using the Bayesian Optimization Toolbox (Nogueira, 2014) for 300 time steps. Each energy hyperparameter was tuned on a small training or validation set by recording logits and finding the combination of energy functions that maximized accuracy.

For experiments on ImageNet, CIFAR10, CIFAR100, and STL10, we only used the classification energy for computational efficiency. This setting can be reduced to a single free parameter, which we denote $\alpha_{\text{logit}}$. The coefficient for mean logit is thus $\alpha_{\text{logit}}$ and the coefficient for max logit is $(1 - \alpha_{\text{logit}})$.

Specifically, the $\alpha_{\text{logit}}$ coefficients we found were: $0.59$ for CIFAR10, CIFAR100, and ImageNet, $0.73$ for STL10,and $0.64$ for our method applied with PRLC's classifiers. For segmentation, the $\alpha_{\text{logit}}$ is $0.74$ with a diffusion energy factor of $0.94$ and a segmentation energy factor of $1.12$. For ImageNet, we also found it helpful to include the mean of top-5 logits with a factor of $0.08$.

For diffusion energy, we subsample the time steps to range from $500$ to $1000$ with a stride of $20$. This is primarily for computational efficiency.

## A.4   3D

For our CO3D experiments, we take 10 random videos from each class, and sample 50 random frames from each video. We crop and preprocess the view following the pipeline in (Liu et al., 2023). Like Appendix A.3, we tune the energy hyperparameter with Bayesian Optimization using a 10% subset of the data. The metric optimized is the difference in mean accuracy of the best five ranks and the worst five ranks. We sort the frames by energy and bin them by their respective video frame ranks, and then compute the accuracy of each rank over the videos. The $\alpha_{\text{logit}}$ coefficient found was 1.31.

For Chrominance invariance results, we use Bayesian Optimization to make the search more efficient. We initialize the GP with 10 random samples and then iteratively search over 20 more samples.

For Objaverse-LVIS (Deitke et al., 2022), we render 400 objects at 36 views, corresponding to the upper hemisphere of azimuth and elevation angles at an interval of 30 degrees. Due to some Objaverse-LVIS containing similar labels which confused the models (e.g., orange vs. mandarin orange vs. tangerine, ring vs. wedding ring, etc.), we filtered the dataset. Specifically, we only kept objects with (1) more than 10% of renders classified correctly and (2) a clear winner class (i.e., the frequency of the most common class should be at least 33% more than 2nd most common class). This selects roughly 30% of the objects.. This generates the test set of images with different viewpoints to evaluate on.

Then, to evaluate FMC, we start at each Objaverse render, simulate Zero123 (Liu et al., 2023) generated images at azimuth circles of an interval of 30 degrees at elevation angles of [-60, -30, 0, 30, 60], taking the minimum energy (best) Zero123 generation as the canonical form. Like for CO3D, we rank the frames, sort and bin them, and compute the accuracy for each rank. However, to isolate the effects of Zero123 on FMC, we rank both the baseline and FMC curves by the probability of the ground truth mask as a proxy for the true ranking of viewpoint. We use the same Bayesian Optimization setting as CO3D. The $\alpha_{\text{logit}}$ coefficient found was 0.79.

## A.5   RELATED WORK (CONT.)

**3D Robustness and pose estimation**: Existing approaches for 3D robustness combine multiple views by pooling features across them Fan et al. (2024); Su et al. (2015); Yang & Wang (2019); Wei et al. (2020; 2022); Hamdi et al. (2021); Kanezaki et al. (2018). However, in many vision settings, multiple views or 3D models may not be available. In contrast, our technique only requires one view at test time—we simulate alternate views with a generative model instead. Chen et al. (2020) learns category-level pose estimation using analysis-by-synthesis. This approach is closely related to our approach; however, it is category-specific, whereas our model is category-agnostic. ImageNet3D (Ma et al., 2024) annotates a large dataset of 3D objects with poses and trains NNs for open-set pose estimation in a supervised manner. In contrast, our method is unsupervised.

**OOD detection**: A variety of approaches have been proposed for OOD detection, including energy functions and generative models (Hendrycks & Gimpel, 2016; Liu et al., 2020; Lee et al., 2018; Liang et al., 2017; Graham et al., 2023), but this capability has yet to be harnessed for invariance.

To our knowledge, this work is the first to leverage large-scale generative models in conjunction with Equation (1) to create provably invariant models.

