# OpenReview forum: "Foundation Vision Models are Unsupervised Image Canonicalizers"
_ICLR.cc/2025/Conference — ICLR 2025 Conference Withdrawn Submission_

### Official Review · Reviewer_wapX · 2024-10-26

**Soundness:** 2
**Presentation:** 3
**Contribution:** 2
**Rating:** 3
**Confidence:** 4

**Summary:**

The paper proposes a training-free method for image canonicalization based on pre-trained foundation models. The method is based on the hypothesis that the canonical image will have the minimal energy evaluated using the foundation models. The authors permute the possible augmentation and use the proposed method to evaluate their energy.

**Strengths:**

- This paper proposes a training-free method for image canonicalization using pre-trained foundation model, which is interesting.
- The paper is easy to follow and the idea is easy to understand.

**Weaknesses:**

- This method formulates the image canonicalization problem as an optimization problem, which needs to permutes all possible augmentation/transformation, which is slow and is sample-inefficient.
- Moreover, the method is dependent on human-designed transformations, e.g. color, viewpoint, rotation, etc. It may have limited potential to generalize real-world transformation that are actually very complex.
- No reason or intuition on why picking the mentioned three vision foundation models. Why using them, not other foundation models?
- No ablation on figuring out which foundation model is useful for recovering which transformation. And no ablation on whether it is necessary to use all three foundation models.
- Besides, I am actually not sure whether this method can generalize to any vision foundation models. For example, DINO/DINOv2, which uses contrastive learning for representation learning might not suitable for this task, as they might not be sensitive to transformations/augmentations.
- The authors should provide a deeper understanding on how to choose these foundation models and how to balance the energy they contribute to the final energy function.

**Questions:**

Please see comments above.

---

> ### Author Response · Authors · 2024-11-15
>
> Thank you for the review. We answer your questions/concerns below:
>
> **“The method is dependent on human-designed transformations; may have limited potential to generalize real-world transformations.”**:
>
> **This understanding is incorrect**: our method can be used with any parameterizable transformation, e.g., the latent space of a generative model. It does not have to be a human-designed transformation. We chose intuitive transformations as these are already popular and easy to understand/debug, but in principle, this could be data-driven. **Importantly, this generality is the key strength of our method** compared to classical methods like equivariant nets!
>
> Our viewpoint transformation example shows this comparative advantage. Equivariant nets cannot be used for viewpoint transformations on images. However, viewpoint transforms can be learned by a generative model (like Zero123). Our method can leverage this generative model to deliver equivariance/invariance. In future work, we will apply this technique to more complex and realistic transformations.
>
> We also argue that even the transformations shown in this paper are helpful – e.g., viewpoint changes often appear in robotics settings. This drastically increases the number of demonstrations required to act as data augmentation in applications such as home robotics [1]. This is because there are currently no reliable methods for viewpoint invariance. Our work represents a useful step towards solving this long-standing problem.
>
> Overall, we agree that it is important to model complex transformations in the real world for real-world robustness. If that is the goal, **our method is more well-suited for complex transformations than classical invariance approaches** and is a useful step toward real-world robustness.
>
> **“Optimization is slow and sample-inefficient.”**:
>
> **Our goal is generality**: The main goal of our work was to show that our canonicalization method is a general and training-free method for invariance. This enables us to generalize to a wide range of transformations (as described above) much better than classical invariance methods. As a result, we focused on generality rather than speed in this work. To our knowledge, methods that use only 1 sample do not generalize across tasks, models, and transformations.
>
> **Multiple samples are necessary for generality**: Please also note that there are no other guaranteed invariance methods that are as general and still fast. It is currently unknown if that goal is even possible.
>
> To our knowledge, existing single-sample solutions for invariance/equivariance require an equivariant/invariant network in the pipeline. This is a consequence of Equation 2 in [3]: Either the canonicalizer or the classifier has to be equivariant. This can be achieved either with an equivariant network (which enables you to sample 1 point but is limited to simple human-designed transforms) or by using multiple samples (which we do). If generality is required, then the only known method so far is optimization.
>
> Please let us know if you disagree and have an example of a method that satisfies the three criteria of generality, invariance, and speed.
>
> **Sample Efficiency**: Even so, for settings requiring many samples (e.g., color shift), Bayesian optimization (BO) is incredibly sample-efficient. For example, in our color shift experiments, 10-30 samples with BO achieve similar accuracy as an exhaustive 21x21 grid (i.e., 441 samples on the grid). This is a nearly **20x speedup**.
>
> Additionally, state-of-the-art “few-shot” Bayesian optimization methods can be even more sample-efficient by learning the statistics of the energy functions [2]. For example, Figure 2 of [2] shows their method achieving the same result in 4-5 samples as naive BO in 20-30 steps. **Thus, there is great potential for speedups in future work.**
>
> For clarity and experimental rigor, we will add a plot of samples vs accuracy for the camera-ready version.

---

> > ### Author Response · Authors · 2024-11-15
> >
> > **“Which foundation models to use, and how to balance them”**:
> > This is a good point, and we will fix it by adding a subsection discussing this to the camera-ready draft. There is a practical and a theoretical answer to your question:
> > - In practice, the best combination of models can be determined by hyperparameter tuning using Bayesian optimization. Practical constraints on computation/memory can be encoded into the objective function to help choose models.
> > - In theory, the probability distribution represented by the energy function must approximate the data distribution at least along the submanifold defined by the transformation. Thus, the requirements for the foundation models are:
> >     - The foundation model must see enough natural data to be able to model the data distribution in a large number of settings.
> >     - The input data distribution to the foundation model should not deviate significantly from the real data, so drastic augmentations (which change the input data statistics a lot) are not allowed.
> >     - The model’s training loss must be able to model the data distribution either explicitly (e.g., flow models, diffusion models) or implicitly (e.g., classifiers through JEM [4]).
> > - One test for whether a model can be used in FMC is whether the model in question can be used for out-of-distribution detection for a wide range of settings.
> > - It would be useful to have a more precise theoretical argument for which models/losses to use. For methods such as DINO/MoCo, we are limited by a lack of deep learning theory to describe their learned distributions. We hope to address this in future work once there is a better understanding of these models.
> >
> > **“Energy function ablations”**:
> > Thank you for the suggestion. We will add the ablations in the appendix section in the camera-ready draft.
> >
> > **“Besides, I am actually not sure whether this method can generalize to any vision foundation models.”**:
> > To clarify, we don’t claim this. While we generalize across downstream models and while our framework is flexible to incorporate new foundation models, we don’t claim that any foundation model would work well. We will add the above clarifications to describe what foundation models should be used in FMC. Please let us know if we misinterpreted your point.
> >
> > **In summary**, we will add (1) a plot of samples vs accuracy, (2) a discussion about foundation model choice, and (3) ablations to the camera-ready draft.
> >
> > Please let us know if you have any more questions. We would also appreciate knowing what you would like to see to change your rating.
> >
> > [1] https://arxiv.org/pdf/2311.16098
> >
> > [2] https://proceedings.mlr.press/v151/tighineanu22a/tighineanu22a.pdf
> >
> > [3] https://arxiv.org/pdf/2211.06489
> >
> > [4] https://arxiv.org/pdf/1912.03263

---

> > > ### Comment · Reviewer_wapX · 2024-11-18
> > >
> > > Thanks for reply. I have some followup questions.
> > >
> > > 1. Regarding "our method can be used with any parameterizable transformation". The authors should provide experimental results on real-world natural images with complex transformations. Now there is no support for your claims.
> > >
> > > 2. Regarding “Optimization is slow and sample-inefficient”. The authors should provide quantitative evaluation on the sample efficiency, on diverse level on transformations that applies, and compare with baselines. The authors provided a study on color shift experiments. However, this involves only one transformation. If I understand it correctly, the theoretical complexity of the method is actually O(T^N), where N is the number of transformation that involves, and T is the number of possible solutions for each augmentation you want to study. I believe this is sample inefficient.
> > >
> > > 3.  Regarding “Which foundation models to use, and how to balance them”. The authors should update the paper to include the discussion right now.
> > >
> > > 4. Regarding “Energy function ablations”. The authors should solve the problem during the rebuttal period.

---

> > > > ### Author Response · Authors · 2024-11-20
> > > >
> > > > **"The authors should update the paper to include the discussion right now.":**
> > > >
> > > > We will do this in the next day or two.
> > > >
> > > > **"Regarding “Energy function ablations". The authors should solve the problem during the rebuttal period.":**
> > > >
> > > > It is difficult to try every combination in this short time, but we will try our best.
> > > >
> > > > **"The authors should provide quantitative evaluation on the sample efficiency, on diverse level on transformations that applies, and compare with baselines.":**
> > > >
> > > > We will also try to run that evaluation. However, in the interest of time, please tell us precisely which setting (transform+model+settings+levels+baselines) you want us to run.
> > > >
> > > > **“The theoretical complexity of the method is actually O(T^N).”:**
> > > >
> > > > Thanks for your question. While your estimate of O(T^N) is correct for **brute-force search** in N dimensions, **we would not recommend brute-force search** in such settings.
> > > >
> > > > We recommend **Bayesian Optimization (BO)** (Sec 4.3), which is significantly more sample-efficient. For example, color actually constitutes a 2D optimization problem (i.e., color ratios R/G and B/G). A brute-force method would require sampling 21^2=441 combinations (T=21, N=2), but with BO, we achieve comparable results with just 10–30 samples (Fig. 5).
> > > >
> > > > **Adding more transformations**—such as adding image contrast (making it 3D) or including affine transforms (totaling 9D)—scale the problem to **general N-dimensional optimization**. An extreme example of this is the latent space of a generative model (e.g., 128D). BO and gradient descent (especially for high dimensions) have been effectively used for optimization in latent space [1,2,3,4].
> > > >
> > > > Therefore, we argue that **N-dimensional optimization is feasible** and can be done with much more sample efficiency than brute-force. These methods may not be guaranteed to find a global minimum (unlike brute-force search) but still achieve state-of-the-art results in practice.
> > > >
> > > > We hope the answer clarifies how we are overcoming the high cost of brute-force search for a practical solution.
> > > >
> > > > \[1\]: https://arxiv.org/abs/2201.11872 \
> > > > \[2\]: https://arxiv.org/abs/1610.02415 \
> > > > \[3\]: https://www.nature.com/articles/s42256-022-00532-1 \
> > > > \[4\]: https://arxiv.org/abs/2006.09191

---

> > > > > ### Author Response · Authors · 2024-11-20
> > > > >
> > > > > **"Regarding "our method can be used with any parameterizable transformation". The authors should provide experimental results on real-world natural images with complex transformations. Now there is no support for your claims.":**
> > > > >
> > > > > We disagree with your point. To clarify our position:
> > > > >
> > > > > 1. Our paper initially claimed to generalize across transformations like rotation, lighting, and viewpoint/3D rotations. We claim that (1) these appear naturally in settings like robotics and (2) that current invariance methods cannot handle difficult transformations like 3D/viewpoint.
> > > > >
> > > > > 2. Your initial review claims that our method (1) depends on human-designed transformations and (2) cannot generalize to complex transformations.
> > > > >
> > > > >    To our knowledge, you have not provided any definition of “complex transformations,” provided any examples of such transformations, or provided any logical arguments to support these claims.
> > > > >
> > > > >    We would appreciate a clear example of what you consider a sufficiently complex transformation and a logical argument to support your claim that our approach cannot handle it.
> > > > >
> > > > > 3. To clarify our previous response, here is the logical argument and its evidence:
> > > > >
> > > > >    -  a. Our method takes advantage of the emerging capability of foundation models. We argue that (1) many real-world complex transformations can be modeled using generative models, and (2) in principle, we can optimize any parameterizable transformation, including the latent space of such a generative model.
> > > > >
> > > > >       Recent advances in generative models have made this a plausible mechanism for handling complex transformations. We concluded that our method is more well-suited to complex transformations than classical methods.
> > > > >
> > > > >    - b. Our **evidence** to support the logical argument is the viewpoint/3D experiments. Viewpoint/3D rotation on images is both complex/difficult and natural.
> > > > >
> > > > >      **Viewpoint/3D rotation on images is, in fact, difficult** because occlusions destroy information in the original image. Existing invariance methods like equivariant nets cannot deal with such a transformation. Thus, a generative model like Zero123 is necessary to synthesize new views, where its latent space is conditioned on the view angles. **Our 3D experiments are an example of optimization in the latent space of a generative model.**
> > > > >
> > > > > Thus, we have proposed a mechanism using generative models and supported it with at least one setting (3D) where we use the generative model. Thus, **we disagree with your assertion that there is no support for the claim.**
> > > > >
> > > > > Also, we are not claiming theoretical completeness for all possible transforms. We do show that our approach handles a broad range of transforms that occur in real-world computer vision domains (more than shown in prior work). Going from simple 2D rotations to viewpoint/3D rotation on images (which previous invariance literature cannot) is already a big step and shows the potential of our method.
> > > > >
> > > > > Finally, we argue that a single conference paper should not have to solve the most general possible version of a 70+-year-old open problem. Our method is a step toward real-world robustness, and we thus argue it is a valuable contribution in its current state.
> > > > >
> > > > > We hope this response alleviates some of your concerns. We look forward to hearing from you with any follow-up questions/concerns.

---

> > > > > > ### Author Response · Authors · 2024-11-25
> > > > > > **Updates and Results**
> > > > > >
> > > > > > We would like to share some updates and results:
> > > > > >
> > > > > > 1. We added a discussion about model choice to the main draft.
> > > > > >
> > > > > > 2. We also added newer 3D viewpoint results to the main draft (Fig 6c).
> > > > > >
> > > > > > 3. Here is a preliminary sample complexity benchmark that you requested: https://imgur.com/a/EcG6olC . As mentioned earlier, our method uses significantly fewer samples than brute-force search. We will run more comparisons for the camera-ready draft. If there is a specific setting you’d like, please let us know.
> > > > > >
> > > > > > 4. To further support our claim about real-world transformations, we took a real-world video from CO3D (see example frames here https://imgur.com/a/ar5pAUK ) and applied our method. Specifically, we segmented the target object with SAM2 and compared the average accuracy over frames with CLIP and OVSeg.
> > > > > >
> > > > > > | Model               | Average accuracy over 98 frames |
> > > > > > |---------------------|---------------------------------|
> > > > > > | CLIP                | 38.8%                           |
> > > > > > | SAM2 + OVSeg        | 51.0%                           |
> > > > > > | SAM2 + Ours + OVSeg | 63.3% (+12.3)                   |
> > > > > >
> > > > > > As seen here, our method can significantly increase accuracy over CLIP and OVSeg for a real-world video. We are now running more examples and further analyses.
> > > > > >
> > > > > > We hope these results address some of your concerns. If there is a more specific example/transformation/setting you would like to see in the next few days, please let us know.

---

> ### Author Response · Authors · 2024-12-03
> **Further updates/results**
>
> We have further updates/results to share: new applications with **real-world transforms** and **ablations**.
>
> As promised, here are some **energy function ablation results** (for segmentation results specifically). We will do further ablations for the camera-ready draft.
>
> | Method       | Pose Accuracy | Avg. Pose Error (degrees) |
> |--------------|---------------|---------------------------|
> | Only CLIP    | 68.9%         | 37.1°                     |
> | Only diff    | 82.7%         | 22.6°                     |
> | Only seg     | 60.4%         | 47.5°                     |
> | seg+clip     | 77.8%         | 26.6°                     |
> | diff+clip    | 89.5%         | 13.5°                     |
> | diff+seg     | 84.8%         | 19.6°                     |
> | seg+diff+clip| 90.6%         | 12.1°                     |
>
>
> **Day-Night transformation**: Since you mentioned wanting to see realistic transformations, we tried day-night transformations recently modeled by a state-of-the-art relighting paper (https://github.com/zhihao-lin/urbanir). We took their trained model checkpoint and applied it to 2000 images from their provided dataset, creating 2000 day-night pairs. Our energy function picks day images **91**% of the time, showing its strong canonicalization ability.
>
> We further plotted the energy function while interpolating between the day and night images in Stable Diffusion 2’s latent space and found that the model energy is consistently lower for day images than night images. Finally, we ran optimization in this restricted subspace of the diffusion latent space and found that our method is indeed able to canonicalize these images. See results here: https://imgur.com/a/mhCjfnb
>
> **Active embodied vision with 6-degree camera movement (x,y,z,phi,theta,roll)**: As another example of more realistic transformations, we applied our energy function to optimizing camera parameters in a virtual environment (modeled by a Gaussian splat, using https://github.com/nerfstudio-project/gsplat). We believe this setting represents a more realistic version of our 3D viewpoint experiments and has useful implications for embodied AI.
>
> Here, a camera moves around a virtual scene, searching for the view that optimizes the energy function. We find that this leads the camera to focus on salient objects in the scene and replicate typical angles (like in our previous experiments). Here are some results: https://imgur.com/a/cnizqMx.
>
> While these results are still preliminary, we hope they show the strength of our approach in dealing with many different kinds of transformations, both analytical and realistic.
>
> We also note that contemporary invariance works only show results on simple transformations like 2D image rotation, so we have already gone above and beyond the standards of this subfield.
>
> If you have any other concerns, please let us know.

---

> ### Author Response · Authors · 2024-12-03
> **Overall Summary of our rebuttal**
>
> During this rebuttal process, we have:
>
> 1. Added a subsection to the paper discussing foundation model choice.
> 2. Provided sample complexity results for the color setting, demonstrating a significant advantage compared to brute-force.
> 3. Provided ablations for the energy functions on segmentation.
> 4. Shown preliminary results on **3 different realistic transformations**: CO3D viewpoint shifts, day-night, and embodied vision in a virtual environment. Our method demonstrates promising canonicalization performance in these experiments, and we plan to do large-scale testing for the camera-ready draft.
>
> We would like to repeat our main point that our approach is significantly more general than contemporary work (e.g., equivariant nets), which shows results on group transformations like 2D image rotations. **Our work not only beats existing baselines (e.g., PRLC) published in similar conferences (NeurIPS 2023) on their own benchmarks but also generalizes to entirely new domains** (e.g., Objaverse in the main draft, plus 3 new settings as shown in the rebuttal).
>
> We hope these results help convey the potential of our approach as a step towards more general invariance methods.

---

### Official Review · Reviewer_CgAq · 2024-10-28

**Soundness:** 3
**Presentation:** 3
**Contribution:** 3
**Rating:** 6
**Confidence:** 2

**Summary:**

FMC introduces a training-free approach to equip foundation models, like CLIP and SAM, with canonicalization-based invariance, enhancing their adaptability across various models and a wide range of downstream transformations.

**Strengths:**

1. This energy function-based training-free method is technically sound and novel.
2. The experiments are convincing and sufficient, validating the proposed methods across multiple dimensions, such as datasets (Sec. 5.1), models (Sec. 5.2), and transforms (Sec. 5.3).
3. The paper is well-motivated (a training-free general-invariance method) and easy to understand.

**Weaknesses:**

Please see the "Questions" section.

**Questions:**

For Takeaway #9,

1. The performance gains in Fig. 6 (a) and Fig. 6 (c) are not monotonous and I would like to see more analysis regarding it.

2.  Zero123 is trained on Objaverse, how about the performance comparison on other datasets, such as OmniObject3D [1] and Co3D.

3.  How about changing the Zero123 baseline to the advanced image-to-3D methods, like One-2-3-45 [2] and Unique3D [3].

[1] OmniObject3D: Large-Vocabulary 3D Object Dataset for Realistic Perception, Reconstruction and Generation. CVPR 2023

[2] One-2-3-45: Any Single Image to 3D Mesh in 45 Seconds without Per-Shape Optimization. NeurIPS 2024

[3] Unique3D: High-Quality and Efficient 3D Mesh Generation from a Single Image. NeurIPS 2024

---

> ### Author Response · Authors · 2024-11-15
>
> Thanks for the review! We are glad that you find our approach to be novel, well-motivated, and convincingly evaluated across different settings. We hope this approach is a useful step towards general canonicalization-based invariance.
>
> As for your questions:
>
> **Fig 6 (a) and (c)**: In case you are referring to the roughness of those curves, that is an artifact of the random sampling used to compute the accuracy statistics. To fix this, we will add error bars and average over more samples for the camera-ready version. Please let us know if we misinterpreted your comment.
>
> **Zero123 + Objaverse**: You are correct in pointing out that Zero123 is trained on Objaverse. Since our main contribution is the energy function for canonicalization, Zero123 is just another augmenter from our perspective (like the 2D rotation or color transform functions). We used Zero123 here, but in principle, our FMC framework could use any 3D viewpoint generation method.
>
> **Better augmenters**: Replacing Zero123 with augmenters like One-2-3-45 would likely lead to much better results if the generation quality is improved. We simulated what the “ideal” generator could achieve by using the ground-truth Objaverse renders. In this case, the improvement is significant as shown in the following figure (with updated data processing; see “updated 3D results” below for details):
>
> https://imgur.com/a/AUDGaXl
>
> **CO3D**: Figure 6 (a-b) shows that our energy function can rank CO3D frames well. In practice, we found it difficult to generate high-quality CO3D 3D generations with Zero123. We were held back by practical issues like unreliable foreground-background separation, frames where the object is not fully in the frame, frames where multiple objects are in the scene, and aspect ratio.
>
> We show an example below. The original image (top) does not contain the entire couch, and the background removed version (bottom) segmented the incorrect object. This limits our ability to use Zero123.
>
> https://imgur.com/a/sKkKjrz
>
> Solving these practical issues is possible but outside the scope of this paper. In future work, we plan to focus more on the 3D setting and try better generative models with real-world datasets, especially for robotics applications like home robotics [1].
>
> **Updated 3D Results**: We noticed that Objaverse had poor label quality. The LVIS labels that came with the Objaverse-LVIS subset have many similar labels which confused the models (e.g., orange vs. mandarin orange vs. tangerine, ring vs. wedding ring, etc.). This was impacting both the ground-truth baseline curve and our curve. By simply filtering the data, we achieved much better results.
>
> We only kept objects with (1) more than 10% of renders classified correctly and (2) a clear winner class (i.e., the frequency of the most common class should be at least 33% more than 2nd most common class). This selects roughly 30% of the objects. Please note that these are preliminary results, and we will add error bars and more points in the plot for our camera-ready draft.
>
> See below for the updated plot, with the new lines in solid and the previous lines in dashed lines:
>
> https://imgur.com/a/99SY0A8
>
> Please let us know if there is anything else we can provide to help increase your rating/confidence.
>
> [1] https://arxiv.org/abs/2311.16098

---

### Official Review · Reviewer_dVF9 · 2024-11-02

**Soundness:** 3
**Presentation:** 4
**Contribution:** 2
**Rating:** 6
**Confidence:** 3

**Summary:**

The paper studies to what extent large pre-trained models can be used to canonicalize images with respect to transformations such as rotation and lighting. The proposed approach is to define an energy $E(I)$ for every image $I$ by combining pre-trained CLIP, stable diffusion and SAM and to use brute-force search or bayesian optimization to find the transformation $t$ that minimizes $E(t(I))$. The finding is that this works quite well, i.e. that the proposed energy can be used to canonize images to get improved downstream performance.

**Strengths:**

1. The approach is simple and novel.
2. The paper demonstrates that CLIP, Stable diffusion and SAM (at least when combined) have a strong knowledge of the distribution of internet images under common image transformations.

**Weaknesses:**

1. The approach needs to apply three large pre-trained models (including 500 steps of stable diffusion) to many transformations of the input image. It must be quite computationally expensive, but this is not commented on.
2. There is no comparison to test-time-augmentation, which would be the most common approach to get invariance/equivariance from a non-invariant/equivariant model. Since the proposed approach requires evaluating pre-trained models on several input transformations it seems not to have a computational advantage over test-time-augmentation, which previous work on canonicalization might have had.
3. The approach is claimed to be training-free, but the energy hyperparameters need to be tuned using bayesian optimization (Appendix A.3).

Post-rebuttal:
- The paper is now more complete, with detailed discussion on computational tradeoffs and comparison with TTA.
- I am not convinced that the approach is practically useful. But the paper can be viewed as a reasonable step towards a practically useful method for canonicalization.

**Questions:**

1. How large is the computational cost? In particular compared to using test-time-augmentation.
2. Is the performance better than using test-time-augmentation?
3. How computationally expensive is the tuning of the energy hyperparameters?
4. What is the typical number of transformations used to find the canonical one? How much does the performance improve for, say, each doubling of the number of transformations?
5. What is the advantage of the proposed approach over using the downstream model for the task at hand for canonicalization? For instance, in image classification, the classification model itself could be used to define an energy similar to the CLIP-energy. (This, again, would be a sort of test-time-augmentation.)

---

> ### Author Response · Authors · 2024-11-15
>
> Thanks for your review. We are glad you appreciate the simplicity and novelty of our approach.
>
> In our opinion, this method can achieve guaranteed invariance/equivariance in much more general settings, allowing us to solve problems that previous approaches like equivariant nets could not. Rather than training new canonicalizers for every new setting, FMC simply extracts the information already encoded in foundation models.
>
> **Accuracy comparison to test-time-augmentation (TTA)**: We agree that comparisons against TTA are helpful. Here are the results for CLIP with C8 rotations. **We will also add TTA as a baseline for every experiment in the paper for the camera-ready draft**.
>
> |               | CIFAR10    | CIFAR100   | STL10     |
> |-----------------|--------------|--------------|-------------|
> | No uprighting | 65.4       | 50.6       | 93.4      |
> | Ours          | 93.7       | 76.2       | 97.5      |
> | TTA           | 82.8 (-10.9)| 61.7 (-14.5)| 96.6 (-0.9) |
>
>
> Result: We achieve significantly higher accuracy than TTA for the same evaluated points. Intuitively, this happens because we choose the best point rather than rely on group averages (which could be dragged down by bad samples). While TTA works well for commonly used small ranges (e.g., 10-degree rotations), it degrades when averaging over larger ranges (i.e., full 360-degree circle) necessary for full invariance.
>
> **Advantage against downstream model energy**: Using the energy function from the downstream model is a special case of FMC if the downstream model is a large model like CLIP.
>
> If the downstream is a smaller model like ResNet50, then there are two main drawbacks: (1) R50’s energy function may not work well outside its training setting (e.g., new datasets), (2) it may not be as accurate as FMC due to seeing less data during training.
>
> To show the difference, we used PRLC’s trained ResNet50 classifier and compared it against our approach.
>
> |               | ResNet50 (CIFAR100) |
> |---------------|---------------------|
> | No uprighting | 69.7                |
> | Ours          | 82.2                |
> | ResNet50 Energy | 77.6 (-4.6)       |
>
> Result: Even in the best-case scenario where the classifier is trained for the specific dataset, its energy function is worse. If the test dataset were different, the gap would likely be even larger (similar to PRLC canonicalizers failing on new datasets in Figure 7).
>
> The lower accuracy combined with the fact that this approach might not work outside its training setting are significant challenges for this approach.
>
> For improved experimental rigor, we will add another table to the appendix with comparisons between FMC and classifier energies.
>
>
> **Computational expense**:  Speed/cost is not our priority since we focused on solving a problem that previous approaches could not solve: a general and training-free method for invariance. With FMC, we propose a method that enables a single canonicalization system to generalize across tasks, models, and transformations.
>
> You are correct that the main paper does not discuss the computational expense. We only briefly mention it in the limitations.
>
> **Our cost** is (# of transforms evaluated) X (Cost of transforming + Cost of evaluating [CLIP + SAM + Diffusion model] + Cost of inference)
>
> **Relative to TTA**, this cost is 1 + (Cost of evaluating [CLIP + SAM + Diffusion model])/(Cost of transforming + Cost of inference)
>
> In practice, when using CLIP as the downstream model and 10 diffusion steps (see “Speedup methods” for details), TTA is roughly an order of magnitude more expensive than simple inference, and our method can be roughly an order of magnitude more expensive than TTA.
>
> This difference is much smaller in 3D settings because the generation cost (from Zero123) dominates energy function costs.
>
> We will add a table detailing the computational cost (in FLOPs) and runtime for each experiment (in seconds) for the camera-ready appendix.

---

> ### Author Response · Authors · 2024-11-15
>
> **Hyperparameter tuning**: We argue that hyperparameter tuning is not comparable to training neural networks. (1) The hyperparameter tuning cost is comparatively negligible because hyperparameters can be tuned on just a few dozen images. There are also very few numbers that need to be selected. (2) Energy hyperparameters don’t suffer the same limitations as trained neural nets since hyperparameters transfer well across different datasets and downstream networks. We will include hyperparameter transfer and sensitivity analyses in the appendix for the camera-ready draft.
>
> **Typical number of transformations required**: For C4 and C8 experiments, we chose 4 and 8 samples, respectively. For viewpoint, we use 60. For our most sample-intensive setting, i.e., color shift experiments with CIFAR100, an exhaustive search requires 21x21=441 samples. We use 10-30 samples in the Bayesian optimization (BO). Doubling (i.e., 20-60) increases the accuracy by **less than half a percent**. Similarly, halving (i.e., 5-15 samples) decreases the accuracy by **less than a percent**.
>
> Since our goal was not computational efficiency, we did not try to decrease the number of transforms in any of these settings. However, in our experience, Bayesian optimization is quite sample-efficient.
>
> Additionally, state-of-the-art “few-shot” Bayesian optimization methods can be even more sample-efficient by learning the statistics of the energy functions [7]. For example, Figure 2 of [7] shows their method achieving the same result in 4-5 samples as naive BO in 20-30 steps. Thus, there is excellent potential for speedups in future work.
>
> We will also add a graph of samples vs accuracy for each of the three transformation settings.
>
>
> **Other Speedup methods**:
>
> **Two-stage pipeline**: We have preliminary results for an approach that may significantly reduce amortized computational complexity by skipping the canonicalization when unnecessary.
>
> For 2D rotations, we can skip the canonicalization if the input image’s energy is significantly lower than nearby poses (+90 and -90 degree rotations). This can detect upright vs. not upright with 95% accuracy.
>
> In summary, with only 3 CLIP inferences, we can detect whether to do canonicalization with 95% accuracy. If the extreme rotations are rare (as in the real world), this can bring significant computational savings and, on average, be even cheaper than TTA.
>
> This approach is similar to the “mental rotation” [8] phenomenon in humans, where we classify familiar poses quickly but go through a slow mental uprighting process to classify unfamiliar poses.
>
> We did not include these results initially since our focus is not computational efficiency, but we hope this alleviates some concerns about the cost of this approach.
>
> **Fewer diffusion steps**: In practice, it is sufficient to use only 10 diffusion steps for evaluation (500th, 550th, 600th, ..., 950th). We use this in our experiments to keep inference costs down. This significantly reduces the overall cost of energy computation. It is also possible to use only one step (500), as shown in [9], but this decreases the accuracy too much. We can include a graph of accuracy vs steps as well.
>
> **Distilling FMC energy into a cheaper (or one-shot) model**: Another computational cost reduction might come from distilling the pose knowledge in these foundation models into a single-shot model, like how [8] distills depth knowledge from diffusion models.
>
> In summary, we will add the following to the camera-ready draft: (1) TTA baseline for every experiment, (2) Downstream classifier energy baseline, (3) computational cost (in FLOPs) and runtime for each experiment (in seconds), (4) Hyperparameter transfer/sensitivity analysis, (5) samples vs. accuracy graphs for bayesian optimization and diffusion steps.
>
> To our knowledge, FMC is the only way to get reliable invariance/equivariance that generalizes across a wide range of settings - outperforming TTA (as shown above). Thus, **we argue that this contribution is valuable even if it is slower than prior approaches since a slow solution is better than no solution**.
>
> Please let us know if you have any questions or if there is something you would like to see to change your rating.
>
> [1] https://arxiv.org/pdf/2206.00051
>
> [2] https://arxiv.org/pdf/2010.11882
>
> [3] https://arxiv.org/pdf/2202.10638
>
> [4] https://arxiv.org/pdf/2309.16672
>
> [5] https://arxiv.org/pdf/2211.06489
>
> [6] https://arxiv.org/abs/2202.00665
>
> [7] https://proceedings.mlr.press/v151/tighineanu22a/tighineanu22a.pdf
>
> [8] https://psycnet.apa.org/record/1971-28060-001
>
> [9] https://arxiv.org/pdf/2303.16203

---

> > ### Comment · Reviewer_dVF9 · 2024-11-17
> >
> > I thank the authors for a well-written rebuttal. The added analysis proposed (1-5) improves the paper's contribution by clarifying the limitations and tradeoffs compared to other approaches. A few comments:
> >
> > 1. The authors say that an argument for not using the downstream classification model for canonicalization is that the test data may be OOD, however in that case the downstream classification model will likely perform poorly even with canonicalization by a better model.
> > 2. Since TTA is an order of magnitude cheaper than FMC, it is not clear that adding more TTA would not close the performance gap. For instance, canonicalization through the downstream classification model followed by TTA with small augmentations close to the found canonical one could be a viable strategy. However, I do think that testing all such variants of TTA can be considered outside of the scope of the present paper.
> >
> > I am willing to raise my score to borderline accept if the authors agree to clearly state the approach's limitations in terms of runtime in the main paper.

---

> > > ### Author Response · Authors · 2024-11-18
> > >
> > > Thanks for your response! We promise to clearly state the runtime limitations in the main draft of the paper and discuss it in more detail using previously mentioned analyses. Since runtime is a key limitation of our approach, we hope this discussion will make the paper more transparent and better overall. Thank you for the suggestion. We also agree with your comments.

---

### Official Review · Reviewer_HfhT · 2024-11-07

**Soundness:** 2
**Presentation:** 2
**Contribution:** 1
**Rating:** 3
**Confidence:** 3

**Summary:**

This paper propose a training-free adaptation approach for vision foundation models, e.g. CLIP and SAM, to adapt on unseen 2D image rotation. They conduct experiments on small datasets e.g. CIFAR and STL10 with C4/C8 rotations. Results show some improvement vs baseline w/o such adaptation.

**Strengths:**

Improvement for robustness of foundation models are useful.  Training free adaptation reduce the limitation of usage of those adaptation.

**Weaknesses:**

In the abstract the author talk about rotation, viewpoint, and lighting. However, in the experiment, they only conduct on 2D image rotation.  The 2D in-plane rotation is far too simple, and easy to solve. One can easy train a additional small estimation via self-supervision (data augmentation in training). Or use some existing approach to detect the rotation first [1]. Unless the author can show some results also for other type of perturbation, otherwise, it is hard to convince me the effectiveness of the approach. Additionally, CIFAR10 and STL10 is consider too small and far from realworld usage, I would command the author to conduct more experiment on larger dataset.

[1] Maji, S., & Bose, S. (2020). Deep image orientation angle detection. arXiv preprint arXiv:2007.06709

**Questions:**

See weakness

---

> ### Author Response · Authors · 2024-11-14
>
> Thank you for taking the time to review our paper. However, we see **critical factual misunderstandings** in your assessment. We point out the critical oversights below. We hope you reconsider your review with an open mind.
>
> **Experiments only on 2D rotation**: Your understanding is incorrect. Figures 5 and 6 show color shift and viewpoint results, respectively. The corresponding experiment settings are described in Section 5.3. Earlier figures/tables focus on 2D rotation because the main baseline (PRLC) evaluates on those, but we do consider viewpoint and lighting (illumination color specifically) as well. We can make these results more prominent in the camera-ready draft.
>
> **Evaluated only on small datasets like CIFAR10/STL**: This understanding is also incorrect. We also evaluate on COCO (Table 2), ImageNet (Takeaway #2 in Section 5.1; Table 3 in Appendix), and subsets of CO3D and Objaverse (Figure 6).
>
> Please also note that popular papers in the invariance learning subfield, e.g.,  Augerino, Prior-Regularized Learned Canonicalization,  LILA [1, 2, 3] in similar conferences (NeurIPS, ICML) have used datasets like CIFAR10/STL rather than larger datasets such as ImageNet. *We still went beyond the standard practices of this subfield to show our claims more rigorously*. We hope you consider this factor in your rating.
>
> We can also make these results more prominent in the camera-ready draft.
>
> **2D plane rotation is too simple**: While it is easy to train a 2D rotation estimator for specialized settings (like our baseline PRLC), our paper addresses a more general approach that works across many settings. One example of why this matters is mobile agents; to enable them to work in the real world, we need a system that can adapt to various transformations and settings.
>
> Specialist approaches like PRLC (our baseline) are insufficient for a generally robust system. PRLC trains specialized networks to estimate rotations on a dataset and uses it to upright the image. However, we find that such approaches (1) fail to generalize to new datasets (Figure 7); (2) don’t work for new transformations by design (e.g., viewpoint, lighting color); and (3) degrade on new downstream models (Figure 9). Put simply, such approaches do not work outside of their training setting.
>
> Our work aims to overcome these fundamental limitations. Notably, we beat these specialized models in all relevant settings *without any training*. This aspect is crucial because it allows us to create *one single approach* that works across many settings. Thus, our work is an important step towards a generally robust system.
>
> Please let us know if we missed something. We would also appreciate knowing specifically what we can do to change your rating of the paper. Thank you for your questions.

---

> > ### Author Response · Authors · 2024-11-25
> > **Please let us know if your concerns have been resolved**
> >
> > Dear reviewer, please let us know if our response has resolved your concerns. Since the discussion period ends soon, it would really help us understand and fix any remaining disagreements or issues. We are eager to hear back from you.

---

### Author Response · Authors · 2024-12-04
**Final Summary**

We thank the reviewers for their engagement and guidance. Here, we provide a summary of the paper’s main contributions, new results from the discussion, and improvements to the paper we will make for the camera-ready version. We request each reviewer to read this summary and consider the changes in the whole context in their recommendations during reviewer AC discussions. In particular, we believe the new and updated results in 3D, day-night transformations, and active vision make the paper much stronger.

**Main Contribution**: Our paper provides an unsupervised approach to achieving invariance. Rather than imposing specific transformations or ranges on the training data or architecture, we build a system of modules to reason about the data distribution. Specifically, we combine the knowledge from foundation models (which, by themselves, do not have invariance) together in a way that enables us to perform canonicalization through unsupervised energy functions. This flexible design enables us to achieve superior levels of invariance and generalize across tasks, models, and transformations without any training.

**Key points**:

**1.** In our initial draft, we already show the capabilities of our approach in more settings than popular NeurIPS and ICML papers that study canonicalization  (e.g., Learned Canonicalization and Prior-Regularized Learned Canonicalization [1, 2]):

   **1a:** We extend our evaluations to ImageNet, whereas prior papers traditionally evaluate on smaller datasets like CIFAR10/STL.

   **1b:** We extend our evaluations on 3D viewpoint changes, a much more complicated transformation that cannot be handled by classic methods like equivariant networks.

**2.** Even so, we further extended our approach to work on day-night transformations and active vision during the rebuttal period, two challenging settings that prior invariance work has not studied (to our knowledge).

**3.** These results show the generalizability of our approach. By extracting the knowledge from foundation models, we generalize (without training) across models, tasks, and transformations.



**Discussion Summary:**

**Reviewer HfHT:** We corrected factual misunderstandings in the review (our paper does look at transformations beyond 2D, and our paper does evaluate on datasets like ImageNet). Unfortunately, HfHT has failed to respond.

**Reviewer dVF9:** The main concerns of this review were (1) runtime, (2) comparison against TTA, and (3) hyperparameter tuning. We provided TTA comparisons showing our advantage, promised to add a subsection detailing runtime cost, and hyperparameter sensitivity (more details below).

dVF9 agrees that our paper is a reasonable step towards a practical canonicalization method in the future and has increased their score in response.

Overall, we believe this discussion has made the paper better and more complete. We are especially grateful to dVF9 for their productive engagement and useful suggestions.

**Reviewer CgAq**: The main concerns of this review were the 3D experiments. We explained the noise in our plots and promised to add error bars. We also showed the potential of our ranking function to perform extremely well with a perfect generator. We also ran preliminary results on CO3D as well as an active vision scenario.


**Reviewer wapX**: The main concerns of this review were (1) the generality of our method, (2) sample complexity, and (3) ablations and discussion about which energy function to use. We fixed  point 3 in the current revision and strongly disagree with their assessment for 1 and 2:

*Generality*: We pointed out that the 3D viewpoint example is already extremely difficult, and traditional approaches like equivariant nets cannot handle it. This means our paper already goes beyond the current invariance methods . Since wapX did not specify which transforms they would like to see, we added several new results on real-world transformations (see results below).

*Sample complexity*: We clarified that we are not using brute-force search and added a sample complexity plot showing significant advantages against the brute-force approach.

To our knowledge, they have not responded yet to these arguments/results.

[1] https://arxiv.org/pdf/2211.06489
[2] https://arxiv.org/pdf/2310.01647

---

> ### Author Response · Authors · 2024-12-04
> **Changes / New Results Made in Rebuttal Period**
>
> See https://anonymous.4open.science/api/repo/ICLR_25_Rebuttal-D1DD/file/index.html?v=fc7f36c7 for a summary of the new results, which we reference in the appropriate sections.
>
> **Real-World Results:**
>
> **Updated 3D Results (Reviewer CgAq)**: We provided updated 3D results on Objaverse-LVIS. We found that Objaverse-LVIS contained many examples of similar labels which confused the models (e.g., orange vs. mandarin orange vs. tangerine, ring vs. wedding ring, etc.). These label quality issues affected both the ground truth and our curve. We filtered out such objects, and our results improved (Fig. 1a in the above link).
>
> Since then, we additionally noticed that due to our data-rendering process, some viewpoints on exact side angles (0, 90, 180, etc.) are aligned front-to-back with the camera, hiding large parts of the object (also called accidental viewpoints) (Fig. 1b in the above link). We changed the input views to the algorithm to be offset by 10 degrees (i.e., choosing non-accidental viewpoints) to avoid this issue and found our results to be further improved (Fig. 1c in the above link).
>
> **Preliminary CO3D example (Reviewer CgAq and wapX)**: We also added a preliminary example on CO3D showing a TV video where our approach achieved a 12.3% improvement compared to a non-invariant baseline. We used SAM2 as the background remover. Critically, we achieved this without any change in our energy function.
>
> **New Day-Night Transformation Results (Reviewer wapX)**: To further evaluate on real-world transformations, we applied FMC on day-night transformations modeled by a SOTA relighting paper (https://github.com/zhihao-lin/urbanir). Our energy function picks day images 91% of the time, and we can canonicalize these images by searching in the restricted diffusion latent space (Fig. 2 in the above link).
> New Active Vision Result (Reviewer wapX): As a “real-world” embodied AI application, we evaluate our approach on an active vision setting to optimize camera parameters (x,y,z,phi,theta,roll) in a virtual environment. We find that our energy function can select camera parameters that focus it on salient objects in the scene at typical angles (Fig. 3 and 4 in the above link).
>
> **Runtime / Cost Concerns:**
>
> *New Sample Complexity Analysis (Reviewer dVF9 and wapX)*: We showed that our Bayesian Optimization approach uses significantly fewer samples than brute force on color (Fig. 5 in the above link).
>
> *New TTA Comparisons (Reviewer dVF9)*: Our approach outperforms TTA when evaluated on the same points (+10.9%, +14.5%, +0.9% on CIFAR10, CIFAR100, and STL10, respectively).
>
> *Runtime Cost vs TTA (Reviewer dVF9)*: We compared the cost of our approach vs. TTA. In practice, TTA is about an order of magnitude greater than plain inference, and our approach is about an order of magnitude greater than TTA.
>
> **Ablations**:
>
> *New Energy Function Ablations (Reviewer wapX)*: We ran ablations on our energy functions for segmentation, showing the benefit of using each foundation model.
>
> *New Downstream Model Energy Comparison (Reviewer dVF9)*: We find that our approach with CLIP outperforms using ResNet50 energy by 4.6%, even on a ResNet50 downstream model.
>
> **Writing**:
>
> *New Discussion on Foundation Model Choices (Reviewer wapX)*: We have added a discussion to the draft on the conditions in which foundation models would be well-suited to use with our approach.

---

> > ### Author Response · Authors · 2024-12-04
> > **Changes / Evaluations for Camera Ready**
> >
> > 1. Highlight color / 3D results earlier in the paper (Reviewer HfHT)
> > 2. Add error bars and expand to more samples in Fig. 6 (Reviewer CgAq).
> > 3. Add new hyperparameter sensitivity analysis to the appendix (Reviewer dVF9)
> > 4. Expand current TTA comparisons to add it as a baseline to every table (Reviewer dVF9)
> > 5. Expand current comparisons with downstream model energy. (Reviewer dVF9)
> > 6. Expand current sample complexity plots and analysis. (Reviewer dVF9 and wapX)
> > 7. Add subsection clearly discussing runtime limitations (FLOPs and runtime) (Reviewer dVF9)
> >
> >
> > Once again, thank you for your time reviewing our paper. We hope we have addressed all of your concerns and hope our responses improve your outlook on our paper.

---

### Note · Authors · 2025-01-23

I have read and agree with the venue's withdrawal policy on behalf of myself and my co-authors.